# Pilot Validation Study of Inertial Measurement Units and Markerless Methods for 3D Neck and Trunk Kinematics during a Simulated Surgery Task

**DOI:** 10.3390/s22218342

**Published:** 2022-10-30

**Authors:** Ce Zhang, Christian Greve, Gijsbertus Jacob Verkerke, Charlotte Christina Roossien, Han Houdijk, Juha M. Hijmans

**Affiliations:** 1Department of Rehabilitation Medicine, University Medical Center Groningen, University of Groningen, Hanzeplein 1, 9713 GZ Groningen, The Netherlands; 2Department of Human Movement Sciences, University Medical Center Groningen, University of Groningen, Hanzeplein 1, 9713 GZ Groningen, The Netherlands; 3Department of Biomechanical Engineering, University of Twente, Drienerlolaan 5, 7522 NB Enschede, The Netherlands

**Keywords:** inertial measurement unit, markerless motion capture, validation study, movement analysis

## Abstract

Surgeons are at high risk for developing musculoskeletal symptoms (MSS), like neck and back pain. Quantitative analysis of 3D neck and trunk movements during surgery can help to develop preventive devices such as exoskeletons. Inertial Measurement Units (IMU) and markerless motion capture methods are allowed in the operating room (OR) and are a good alternative for bulky optoelectronic systems. We aim to validate IMU and markerless methods against an optoelectronic system during a simulated surgery task. Intraclass correlation coefficient (ICC (2,1)), root mean square error (RMSE), range of motion (ROM) difference and Bland–Altman plots were used for evaluating both methods. The IMU-based motion analysis showed good-to-excellent (ICC 0.80–0.97) agreement with the gold standard within 2.3 to 3.9 degrees RMSE accuracy during simulated surgery tasks. The markerless method shows 5.5 to 8.7 degrees RMSE accuracy (ICC 0.31–0.70). Therefore, the IMU method is recommended over the markerless motion capture.

## 1. Introduction

Musculoskeletal symptoms (MSS) are a major health issue in different kinds of occupations [1]. Surgeons are a group of healthcare professionals who are at high risk for developing MSS [2], such as neck pain and back discomfort or pain [3,4,5,6]. Possible physical risk factors of MSS development are (1) prolonged working in the same posture [7], (2) unfavourable working postures [4,7,8], and (3) repetitive movements [9,10].

Quantitative analysis of the working posture of surgeons in the operating room can provide valuable information for ergonomic interventions to reduce the development of MSS among them [11]. In a laboratory setting, optoelectronic motion capture methods are accepted as the gold standard to measure anatomical landmarks and derive body postures and joint angles [12,13]. However, optoelectronic systems are dependent on a stationary laboratory environment and cannot easily be used in real-life working environments such as in the operating room with surgeons wearing a gown.

Alternatively, multiple inertial measurement units (IMUs) can be used to estimate body postures outside laboratories [11]. While IMU-based movement analysis has proven to be valid for joints with a large ROM during gross motor tasks such as knee flexion-extension motion during walking [14], estimating spinal postures in the sagittal and frontal plane during gait [15], assessing position accuracy in static postures [16], and trunk and neck kinematics during small movements [17], their validity for large neck and trunk in multiple planes remains to be established. A disadvantage of IMU-based movement analysis systems is that they require complex data processing steps (e.g., sensor fusion of multiple IMUs) and calibration procedures with the participants standing in a neutral position [18]. These data processing and calibration steps of IMU-based motion capture analyses can be cumbersome and require high technical expertise, limiting their use to biomechanical experts. 

Another alternative for movement analysis in daily life is markerless motion capture, a method to analyse body movement without placement of markers in the body, based on deep learning technology [19]. For example, DeepLabCut [20] and Anipose [21] are open-access pose-estimation toolboxes for markerless motion tracking. DeepLabCut can be used for training a deep learning model and performing 2D postures analyses [22]. Anipose is an addition to DeepLabCut and can be used for estimating 3D postures [23]. While the validity and accuracy of markerless motion capture methods have been established for two-dimensional joint angle analysis in the sagittal plane [24] and 3D position analyses [25,26], none of the previous studies validated markerless motion capture methods on the 3D neck and trunk kinematics. 

This research aimed to establish the accuracy and validity of IMU-based and markerless-based systems (DeepLabCut & Anipose) as compared to optoelectronic motion capture systems in estimating 3D neck and trunk kinematics during simulated surgery tasks.

## 2. Materials and Methods

### 2.1. Participants

Ten healthy subjects (5 males, 5 females; age: 24 ± 3 years; height: 181.9 ± 11.6 cm; mass: 79.1 ± 13.7 kg, body mass index, 23.9 ± 3.6 kg/m^2^) participated in this study. This study was approved by the local Medical Ethics Review Board of the University Medical Center Groningen (METc 2021/385). All participants were informed about the aim of the study and signed written informed consent. The inclusion criterion was: (1) at least 18 years of age. The exclusion criteria were: (1) incapacity to follow instructions, (2) history of medical disorders that may affect movement patterns.

### 2.2. Materials

#### 2.2.1. Marker Motion Capture Measurement

A 10-camera VICON motion capture system with Nexus 2.12.1 software was used to collect bony landmark positions at 100 Hz. The Upper Body Plug-In-Gait model [27] was extended with the following landmarks: left distal clavicle (LCLAV), right distal clavicle (RCLAV) and left back (LBAK) (blue dots in Figure 1). Static calibration and dynamic movement trials of participants were recorded for subsequent kinematic processing.

#### 2.2.2. IMU Motion Capture Measurement

Four XSens MTw inertial measurement unit sensors (Xsens Technologies B.V., Enschede, the Netherlands) containing 3-axis linear accelerometers (range: ±160 m/s^2^), gyroscopes (range: ± 2000 deg/s) and magnetometers (range: ± 1.9 Gauss) were used to track the head and trunk segments. The size of sensor was 47 × 30 × 13 mm and the weight was 16 g. The IMUs were placed on the back of the head, the proximal sternum and the upper spine on the spinous process of T5 and above T10 (orange rectangles in Figure 1) by the double-sided tape. Pilot testing revealed that three IMUs provide sound kinematics and were selected for measuring the trunk segment. The IMU’s positive z-axis points forward, and the positive y-axis points up. The IMUs were calibrated with the participant standing in an erected posture. The Awinda Station connected to the laptop was used for collecting synchronizing wireless data from all MTw sensors via MT manager software (v2021.0.1 build 6752). The orientation data was stored in the laptop after each recording. The sampling rate was 100 Hz.

#### 2.2.3. Markerless Motion Capture Measurement

Figure 2 shows a schematic overview of the experimental set-up. The markerless motion capture system consisted of 4 video cameras (2× Panasonic HC-VX980 (Osaka, Japan) and 2× Sony 4K FDR-AX53 (Tokyo, Japan)) with an image resolution of 1920 × 1080 pixels at a sampling rate of 50 Hz. An audible cue was used to synchronize data recording between cameras. To obtain the intrinsic (focal length) and extrinsic (position and orientation) parameters of each camera, a ChArUco board was used [28]. Calibration was repeated until the reprojection error (geometric error corresponding to the image distance between a projected point and a measured one) was less than 1 pixel [21]. After data acquisition, Shotcut software (https://shotcut.org/ (accessed on 24 April 2021)) was used to synchronize the videos for each participant.

### 2.3. Study Approach

To establish the accuracy and validity of the IMU and markerless motion capture methods for neck and trunk kinematics, the following outcome variables were calculated: (1) 3D neck angles: flexion/extension (FE) angle, lateral bending (LB) angle, transverse angular rotation (AR) angle between head segment and trunk segment at the level of C7 and T1; and (2) 3D trunk angles: flexion/extension (FE) angle, lateral bending (LB) angle, transverse angular rotation (AR) angle in the global coordinate system.

The 3D neck angles and 3D trunk angles obtained by IMUs and the markerless-based system were compared with an optoelectronic marker-based system (gold standard) during two movement tasks:(1) movements in primarily a single anatomical plane (SP-movements, SP is the abbreviation of single anatomical plane) and (2) a simulated surgery task in multiple planes. For SP-movements, each subject was asked to perform neck and trunk flexion and extension, left and right lateral bending and left and right rotation subsequently. 

The simulated surgery task was a task that involved simultaneous and repetitive neck and trunk movement in the combined frontal, sagittal and transverse planes. Our goal was to simulate the working postures and repetitive movements as performed during surgical interventions. To simulate the surgery environment, an OR table was placed in the lab. Participants were instructed to transfer small objects in a box on the table from left to right using forceps in both hands. The details of the tasks were selected through observations during surgical training and informal interviews with surgical training managers of the Skills Center of the University Medical Center Groningen. A detailed description of the tasks is shown in Figure 3 and Table 1. 

Before each trial, the participants were asked to stand up straight, facing forward as a neutral reference position. During both SP-movements and the simulated surgery task, neck and trunk motion were captured with a marker-based optoelectronic system, multiple IMUs and a markerless-based system simultaneously. Each participant performed the series of SP-movements and the simulated surgery task twice. The first trial of each participant’s movement task was used for training the DeepLabCut model and excluded from accuracy and validity analyses. Trunk flexion was used for time synchronization of the motion data obtained by 3 methods based on a cross-correction algorithm [29].

### 2.4. Data Analysis

For all three assessment methods (Section 2.2.1, Section 2.2.2 and Section 2.2.3) the same customized OpenSim 4.3 musculoskeletal model was used [30]. The customized model was modified from the full body thoracolumbar spine model and locked all the degrees of freedom except for: pelvis translation_x, pelvis translation_y, pelvis translation_z, trunk FE, trunk LB, trunk AR, neck FE, neck LB, neck AR) [30]. The model used in this research can be found in the Appendix A. The standard OpenSim scaling and inverse kinematic workflows were used to compute neck and trunk angles from 3D marker data, IMU data and markerless data [31,32]. 

#### 2.4.1. Marker-Based Data Processing

Gaps in the raw optoelectronic motion capture data were filled (Gap-fill Woltering filter, max gap length 10 frames) and filtered using a 4th order Butterworth lowpass filter (6 Hz) to remove high-frequency noise by Nexus 2.12.1 software. The static calibration trial was used for scaling to obtain a customized model of the subject in the OpenSim. The scaling maximum and RMS marker errors of bony landmark positions were less than 1 cm for each participant. Using the scaled model, the OpenSim Inverse Kinematics tool (IK) (OpenSim 4.3) was used to estimate 3D neck and trunk kinematics. The IK maximum errors and RMS errors for bony landmarks were less than 4 cm and 2 cm, respectively in accordance with common practice [33] (OpenSim IK Best Practices https://simtk-confluence.stanford.edu:8443/display/OpenSim/Simulation+with+OpenSim+-+Best+Practices (accessed on 11 November 2021)).

#### 2.4.2. IMU-Based Data Processing

MT manager [34] was used to export IMU data (3-axis acceleration and direction cosine matrix) based on the Xsens fusion filter algorithm. Gap filling was performed when less than 10 consecutive data points were lost. A static calibration trial of the subject standing in a neutral pose was used for calibration. Based on the scaled model and orientation files, customized MATLAB scripts were used to perform the OpenSim IMU Inverse Kinematic workflows. 

#### 2.4.3. Markerless-Based Data Processing

The first trial of each movement task was used for algorithm training. In total, 200 frames were extracted based on DeepLabCut algorithm from four cameras and 10 participants for deep learning model training. The virtual tracking points on the head and trunk (three yellow dots on the head: left and right of the eyebrows, on the chin; and four red dots on the trunk: on the left and right distal clavicle, proximal and distal of the sternum) were labelled manually (shown in Figure 4a). DeepLabCut [20] was used to train the algorithm and predict the 2D position of tracking points on a GPU (GEFORCE RTX 2060 Max-Q, Nvidia Corp, Santa Clara, CA, USA) for 120,000 iterations in around 12 h. Anipose [21] was used to analyse the separate videos based on the trained model and predict the 3D coordinates of tracking points. The workflow was followed as described previously [20,21]. The tracking point position data were further processed with OpenSim software as described in Section 2.4.1. to obtain segment positions and joint kinematics. The workflow for the markerless method is summarized in Figure 4b.

### 2.5. Statistical Analysis

To establish the accuracy and validity of the IMU and markerless computations of neck and trunk kinematics, the following variables were computed and statistically analysed: The root mean square error (RMSE) between the IMU/markerless-based method and the marker-based method of the 3D neck and trunk angles over total movement time.The difference in absolute ROM between the IMU/markerless-based method and marker-based method of the 3D neck and trunk angles. The first data point of the angle-time series from each measurement system was subtracted to correct the offset between systems. For SP-movements, the ROM was defined as the maximum difference between the starting anatomical angle and the maximum angle of the neck and trunk [35]. For simulated surgery tasks, the ROM was defined as the difference between the minimum and maximum angle [17].Relative ROM error, the ratio of IMU/markerless ROM difference to the gold standard ROM.Paired *t*-tests on the mean differences in ROM between IMU/markerless-based method and marker-based method to obtain systematic biases.Bland–Altman plots of the IMU and markerless method for 3D neck and trunk ROM were used to show the limits of agreement and systematic biases.The intraclass correlation coefficient ICC (2, 1) for ROM between the IMU/markerless-based method and marker-based method to establish the validity of the system. ICCs were considered as follows: 0.9–1 as excellent, 0.70–0.89 as good, 0.40–0.69 as acceptable, and <0.40 as low correlation [36]. The level of significance was set at 0.05.

Statistical analyses were performed in IBM SPSS Statistics, version 27. As a quality criterion, measurement errors (RMSE) within 5.0 degrees were interpreted as acceptable [37].

## 3. Results

Representative 3D neck and trunk angles during SP−movements and the simulated surgical tasks are shown in Figure 5 and Figure 6.

### 3.1. Accuracy and Validity for IMU-Based Neck and Trunk Kinematics

The RMSE between the IMU-based method and the marker-based method were all below 3 degrees except for the trunk AR (4.7 degrees) and neck FE (3.7 degrees) when moving in a single plane (SP). During the simulated surgery task, the RMSE was less than 3.0 degrees for the trunk FE & LB, and less than 4.0 degrees for other neck and trunk angles. The ROM differences between the IMU-based method and the optoelectronic system were less than 3 degrees except for the trunk AR (5.0 degrees) and neck FE (5.4 degrees) during SP-movements. The ROM differences between methods during the simulated surgery task were less than 3 degrees (shown in Figure 7 and Figure 8 and Table 2). Limits of agreement (LOA) and mean relative ROM error for the IMU method during SP-movements and the simulated surgery task are shown in Table 2. There were systematic biases for ROM on the trunk FE/LB/AR and neck FE (*p* < 0.05) for SP-movements. The IMU method underestimated trunk FE, trunk LB and trunk AR, but overestimated neck FE angles. During simulated surgery tasks, there were systematic biases for ROM on trunk FE and neck AR (*p* < 0.05). Trunk FE was underestimated, but the neck AR ROM was overestimated.

The ICC (2,1) for ROM between IMU and the optoelectronic measurement system showed excellent correlation (>0.9), except for neck FE where ICC was good [ICC (2,1) = 0.8] during the simulated surgery task (Table 3). 

### 3.2. Accuracy and Validity for Markerless-Based Neck and Trunk Kinematics

The RMSE between the markerless-based method and the marker-based method were all below 10.0 degrees except for the trunk AR (14.9 degrees) and neck AR (15.2 degrees) during SP-movements. For the simulated surgery task, the RMSEs were less than 6 degrees for the trunk FE & LB, and less than 8.0 degrees for the neck FE, LB and AR. 

The markerless-based method had less than 7.0 degrees of ROM difference except for the trunk AR (11.7 degrees) and neck AR (18.5 degrees) during SP-movements. During the simulated surgery tasks, ROMs were less than 5 degrees for the trunk LB, trunk AR and Neck AR, and less than 12 degrees for the trunk FE, neck FE and neck LB (shown in Figure 9 and Figure 10 and Table 2). Limits of agreement (LOA) and relative ROM error for the markerless method during SP-movements and the simulated surgery task are shown in Table 2. There were systematic biases on trunk FE, trunk AR and neck AR (*p* < 0.05) for SP-movements. The markerless method underestimated the ROM for trunk FE, trunk AR and neck AR. During simulated surgery tasks trunk FE and neck LB ROM were systematically underestimated (*p* < 0.05).

ICC (2,1) between markerless and marker-based kinematic computations showed good validity of the trunk FE and neck FE movements and trunk LB ICC was within acceptable ranges during SP-movements (Table 3). The ICCs (2,1) between the markerless and marker-based kinematic computations were acceptable for trunk FE, trunk AR, neck LB and neck AR. ICC (2,1) for trunk LB was good during the simulated surgery task. Table 3 shows the summary of ICCs results.

## 4. Discussion

In this study, we aim to establish the validity and accuracy of the IMU and markerless method against the marker-based method (Vicon). IMU-based computations of 3D neck and trunk angles have good-to-excellent validity with 2–5 degrees RMSE accuracy as compared to optoelectronic motion capture systems during simple and complex neck and trunk movements. The markerless method shows an acceptable-to-good validity to establish trunk kinematics only in the sagittal and frontal plane with RMSEs of 5–10 degrees during both SP-movements and simulated surgery tasks. However, the markerless method shows poor outcomes for neck kinematics. Our results show that IMU-based methods are valid to assess neck and trunk kinematics during simulated surgery tasks but that markerless methods such as DeepLabCut exceed the threshold for acceptable kinematic accuracy [37].

### 4.1. IMU Motion Capture Method

The RMSE and ROM difference values of the IMU method ranged between 1.0 to 5.5 degrees showing good accuracy for estimating neck and trunk kinematics during simulated surgery tasks [37]. The accuracy results from the IMU-based methods are comparable to previous studies reporting RMSEs and ROM for the trunk FE during laparoscopic surgery tasks [17]. A characteristic for laparoscopic surgery tasks is that these activities involve only small movements limiting the generalizability of the findings to other surgery tasks. We add to these findings and show that IMU-based methods maintain a high level of accuracy even during complex simulated open surgery tasks with larger ROM in the neck and trunk segments. In addition, contrary to previous studies comparing optoelectronic systems with IMU measurements we placed the retroreflective markers on anatomical landmarks rather than on the IMU itself. While this methodology led to an increase in RMSE’s and ROM differences we propose that using anatomical landmark definition allows better validation of IMU based motion capture systems for movement analyses [38,39]. The ICC (2,1) results from our studies showed good-to-excellent (0.80–0.99) agreement with the gold standard in multiple planes and were also comparable to or even better than the previous studies (0.63–0.99) [14,17]. The 95% CI for Trunk LB, AR and Neck FE for SP-movements of the IMU method are large, the reason might be the small sample size and the fact that ROM data were not normally distributed. 

During SP movements in four out of six degrees of freedom, ROM differences were significantly different as compared to the gold standard, but only two out of six differed significantly during simulated surgery tasks. This might be explained by the fact that the IMU method has similar relative ROM error in SP-movements and simulated surgery tasks, but SP-movements generally have a larger ROM than the simulated surgical task. The larger ROM may lead to a larger difference compared with the gold standard and lead to a significant difference. For future studies, it is recommended to test more subjects and consider measurements with variability in ROM.

In our study, when assessing movements with IMUs one should especially be careful in interpreting the estimated rotation angles since they contain the largest ROM differences and RMSE as compared to optoelectronic measures. A possible reason for underestimating rotation angles is, that the IMUs are placed on the sternum and spine, while some of the markers are placed more lateral on the clavicula. When rotating the trunk, pro-and re-traction at the shoulder joint might have led to increased estimates of trunk rotation angles with the optoelectrical system. In our study, the trunk segment is considered as one rigid body, future studies might consider using a more complex model to establish the extent to what the use of a limited number of IMUs leads to wrongly estimating the amount of trunk rotation.

As compared to optoelectronic systems the practical advantage of IMU based motion capture in work-related situations is large since there is no need for optical camera systems and marker placement on the participants clothes (e.g., a surgeons’ gown). In the future studies, monitoring kinematics of surgeons in the operating room can provide feedback and input (neck and trunk range of motion, estimated support force) for developing preventive devices such exoskeletons.

### 4.2. Markerless Motion Capture Method

The markerless method shows ROM and RMSE accuracy within 3 to 12 degrees for the neck and trunk kinematics during the simulated surgery task. The accuracy results were comparable to the previous research on 2D kinematics of human walking [21], which reported angle errors of less than 16 degrees in over 90% of frames and less than 10 degrees in over 75% of frames. Our study showed the markerless method had an acceptable-to-good agreement (ICC 0.55–0.83) on trunk kinematics in the sagittal and frontal plane during both SP-movements and simulated surgery tasks. The markerless ICC results are comparable with previous research reporting ICCs ranged from 0.37 to 0.82 during a bilateral squat in the sagittal plane [24]. We add to this finding that the markerless method can also establish similar ICC values in the frontal plane for trunk kinematics during both SP and simulated surgery tasks. 

The markerless method is not stable. Generally, three reasons could exist for the poor kinematics of the markerless method. The first and most important reason is the occlusion of tracking points during the movement. Tracking points can be blocked when the participant reaches the limit of the ROM and can be blocked by their own segment (Figure 11). In the case of occlusion of multiple successive frames, the deep learning model loses accuracy and fails to accurately predict the tracking point position. Inaccurate estimates of tracking points’ positions in turn lead to inaccurate estimates of joint angles and body postures [25]. The second reason is the chosen camera setup. In our setup, the four cameras were mainly positioned in the sagittal and frontal planes (Figure 3). As a consequence, the change of tracking point position in the transverse plane is smaller than in the frontal and sagittal planes, in other words, the markerless system in our setup has a low resolution for transverse plane movements. This setup may magnify the tracking point error in the transverse plane and may lead to larger RMSE and ROM differences for neck and trunk AR. Sufficient cameras to track all key points across possible pose configurations can lead to a better result [21].

The third reason is the tracking error of the DeepLabCut algorithm. To track the head segment, we select the eyebrows and chin as tracking points. The eyebrow and chin may not be obvious features as retroreflective markers. Therefore, it can lead to tracking errors even if there is no occlusion of the head during the movement. In further research, we recommend putting ‘stickers’ on the head to make the tracking points easy to recognise and improve neck kinematic results.

In addition to the main reasons, it should be mentioned that, we had to take out the SD cards from each camera to obtain calibration videos and put them back after calibration. This process may slightly change the orientation and position of the cameras, which can possibly affect triangulation and may cause bad neck and trunk kinematics results. 

### 4.3. Limitations and Recommendation

There were several limitations in this study: (1) the sample size of our pilot study is ten and the participants were all young participants (21~30 years.) and not surgeons, a larger sample size can provide a better estimation on intraclass correlation coefficient, RMSE, LOA and paired *t*-test. Besides, the participants’ movement patterns may differ from surgeons, limiting the generalizability of the systems. It is recommended to perform the test with medical staff in a real OR environment in future research. (2) In this study, we used sound signal for synchronizing the 4 cameras. It might be better to use cameras with built-in synchronization functions to achieve higher synchronization accuracy. (3) Training model: In this study, we trained 200 frames for the deep learning model and these frames may not include rare behaviours. The model can fail to predict the right position of tracking points if there are rare movements in the videos. This may lead to tracking errors. In our preliminary observations, the markerless method can yield good results in 3D kinematics if the tracking errors and tracking points occlusion are minimized. Therefore, a neural network which is trained by more frames including rare movements should give better tracking points detection. It may help to produce more accurate and stable 3D kinematics results. (4) We did not assess long-term motion capture and the influence of magnetic field on IMU based outcomes. Therefore, further research should include trials of longer duration and in magnetic fields similar to ones present at an OR.

## 5. Conclusions

In this study, we validated IMU and markerless motion capture-based methods to measure neck and trunk kinematics against the gold standard optoelectronic motion capture method for movements primarily in a single plane and simulated surgery tasks. The IMU-based method has shown good-to-excellent validity compared to the gold standard for simulated surgery tasks. The present markerless method is not yet sufficiently valid, but it might have the potential for 3D movement analysis in the OR if the camera setup and model training are improved and tested. It is recommended to use IMU for the kinematics analysis of head and trunk motions in the OR.

## Figures and Tables

**Figure 1 sensors-22-08342-f001:**
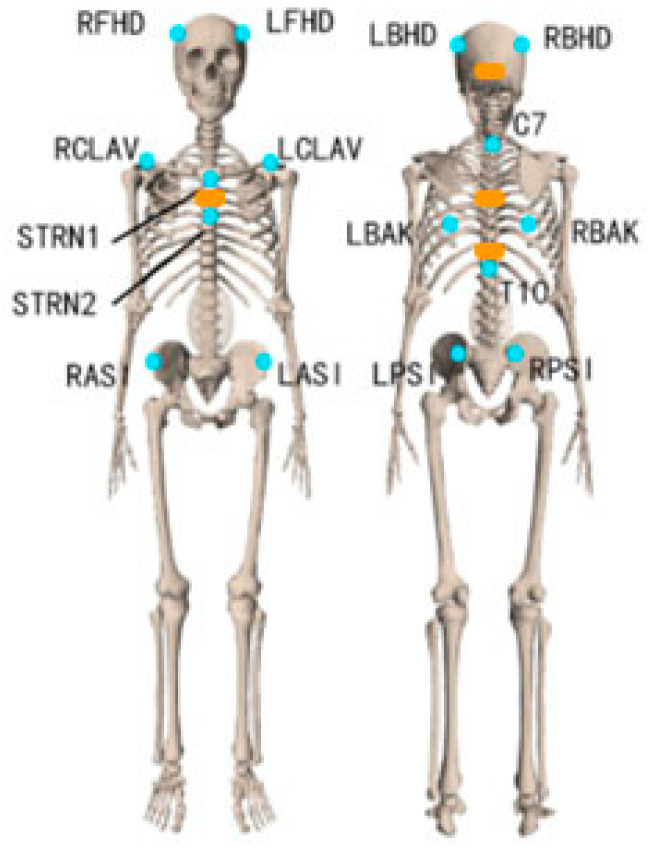
Marker (blue dots) & IMU (orange rectangles) placement.

**Figure 2 sensors-22-08342-f002:**
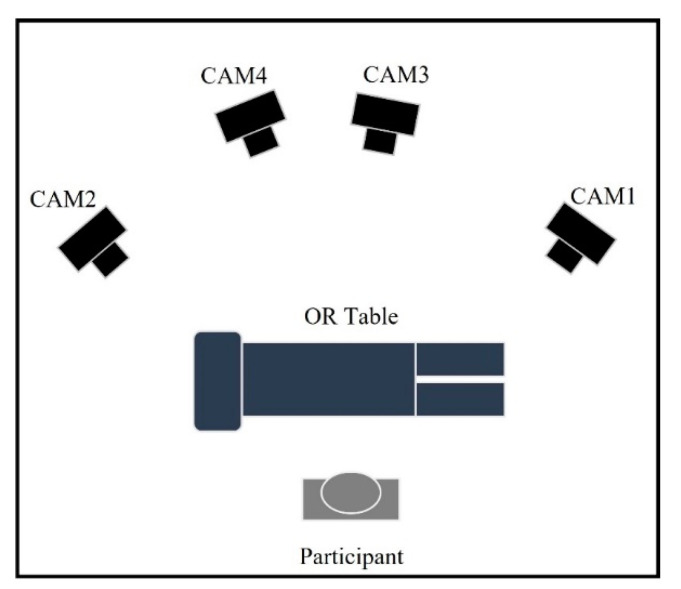
Schematic of Markerless-based motion capture (Top view).

**Figure 3 sensors-22-08342-f003:**
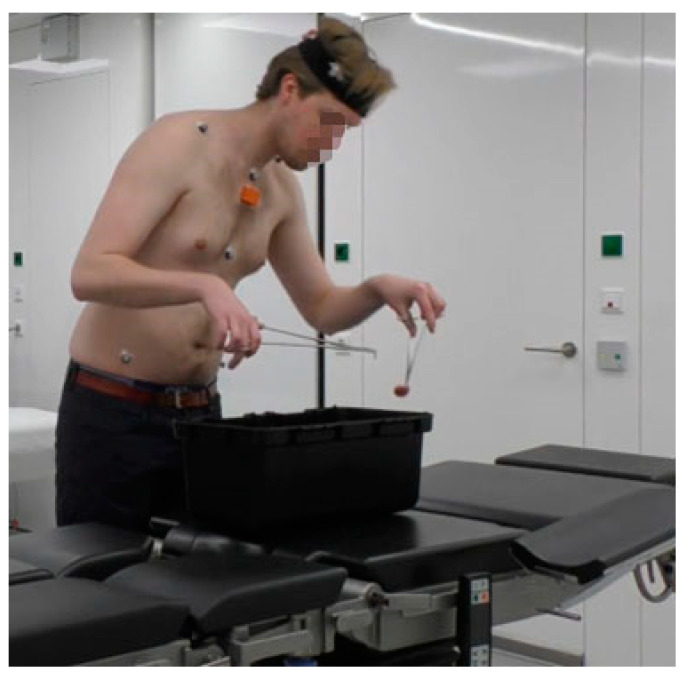
Example of simulated surgical tasks.

**Figure 4 sensors-22-08342-f004:**
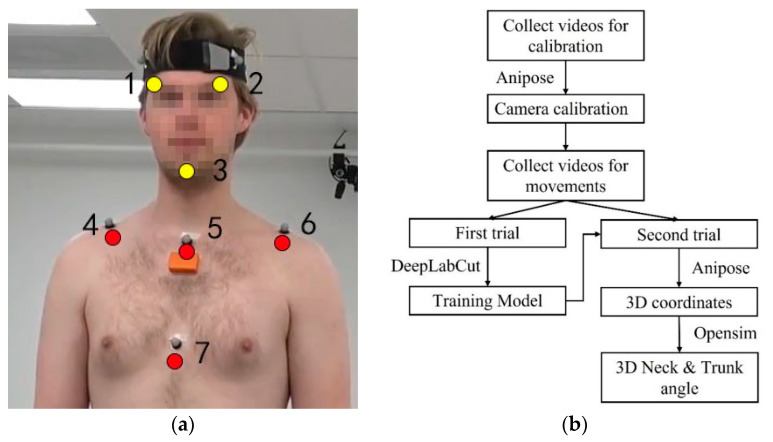
(**a**) Virtual tracking points of the markerless method; (**b**) Workflow of markerless method.

**Figure 5 sensors-22-08342-f005:**
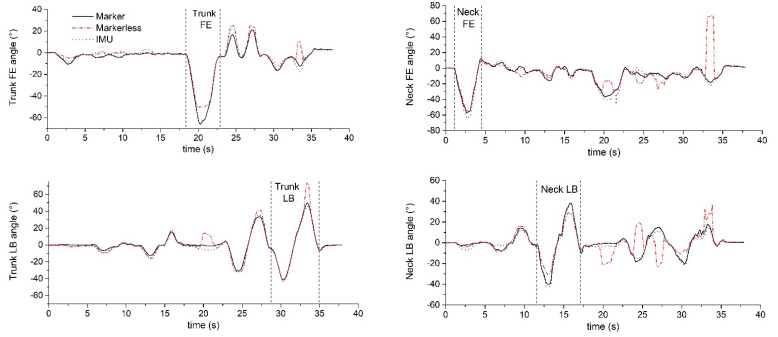
Exemplar SP−movements, the movement sequence is Neck FE, AR, LB and Trunk FE, AR, LB (between to dash line was the simple plane movement).

**Figure 6 sensors-22-08342-f006:**
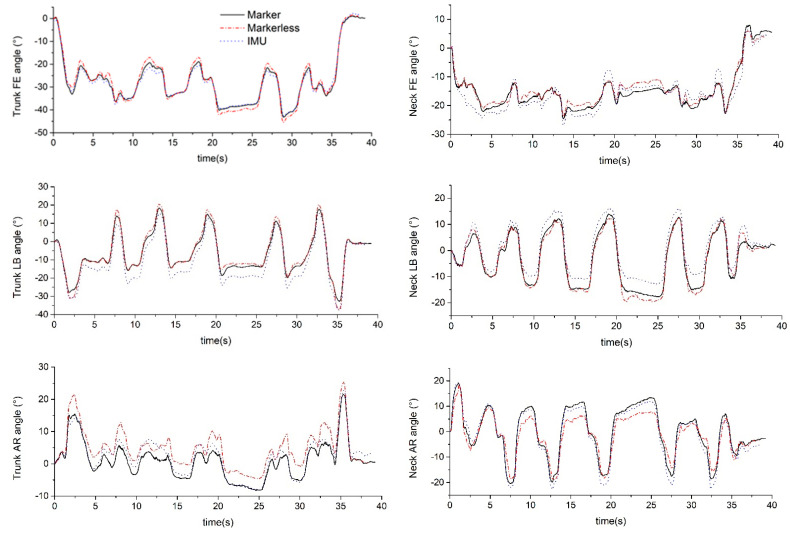
Exemplar simulated surgery task.

**Figure 7 sensors-22-08342-f007:**
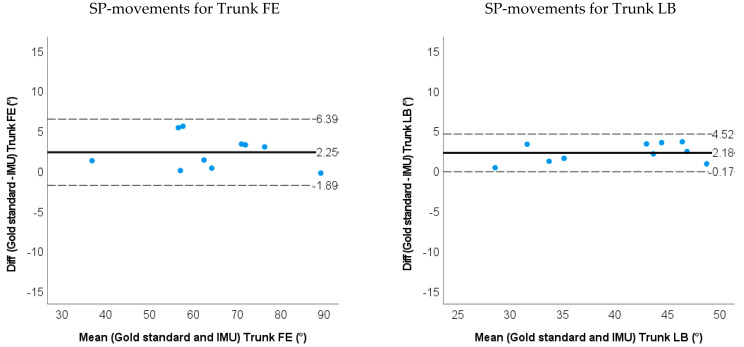
Bland−Altman plots for ROM of the trunk and neck angle obtained by the gold standard and IMU method for SP-movements. The solid line represents the mean difference, and the dashed line represents the 95% limits of agreement.

**Figure 8 sensors-22-08342-f008:**
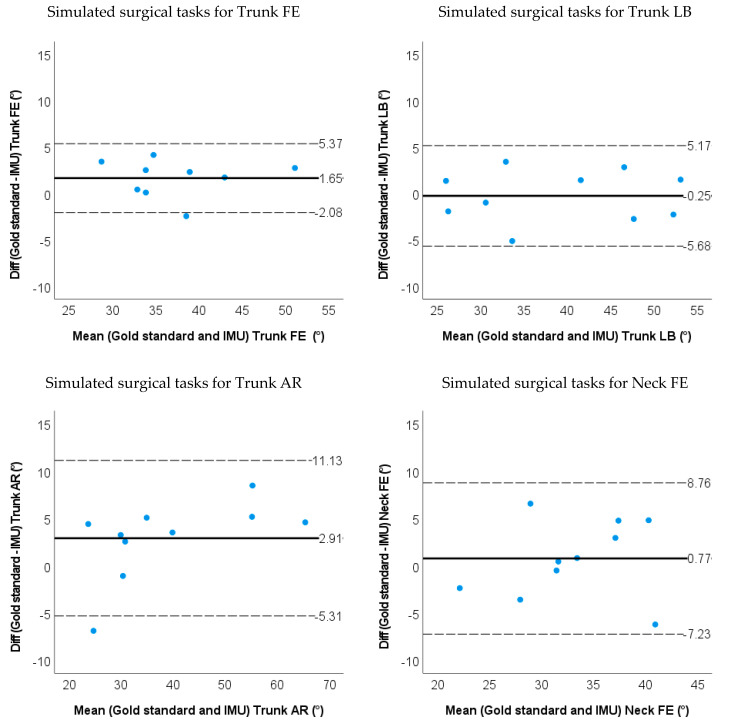
Bland−Altman plots for ROM of the trunk and neck angle obtained by the gold standard and IMU method for the simulated surgical tasks. The solid line represents the mean difference, and the dashed line represents the 95% limits of agreement.

**Figure 9 sensors-22-08342-f009:**
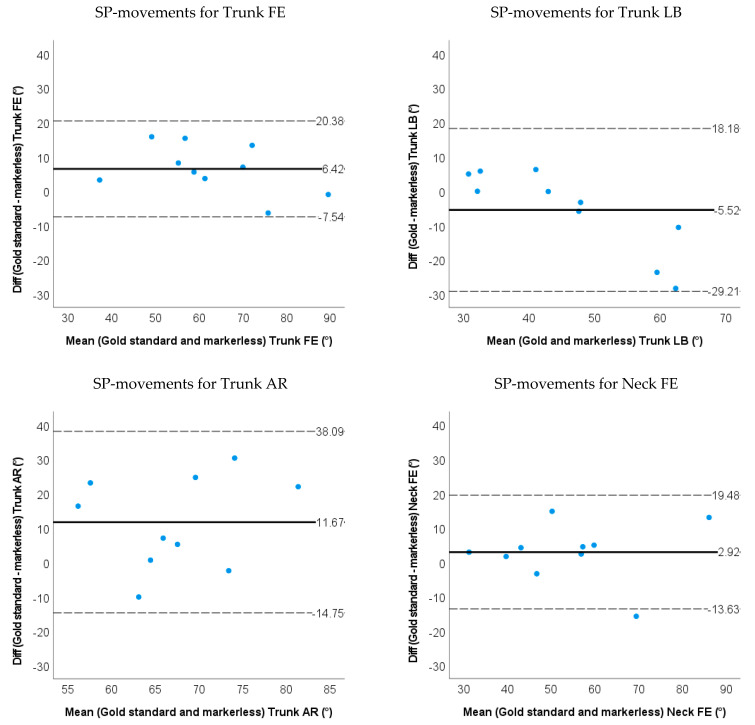
Bland−Altman plots for the ROM of the trunk and neck angle obtained by the gold standard and markerless method for SP-movements. The solid line represents the mean difference, and the dashed line represents the 95% limits of agreement.

**Figure 10 sensors-22-08342-f010:**
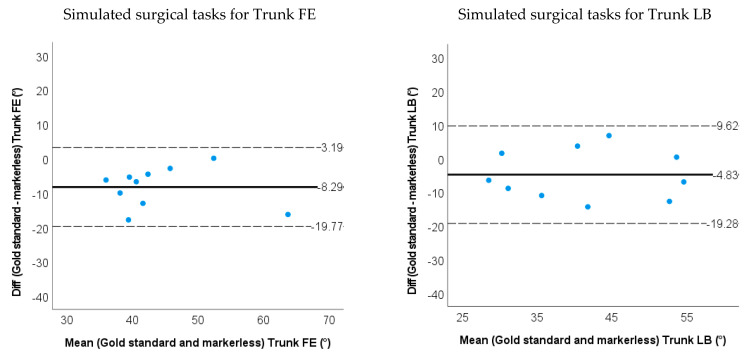
Bland−Altman plots for the ROM of the trunk and neck angle obtained by the gold standard and markerless method for the simulated surgery tasks. The solid line represents the mean difference, and the dashed line represents the 95% limits of agreement.

**Figure 11 sensors-22-08342-f011:**
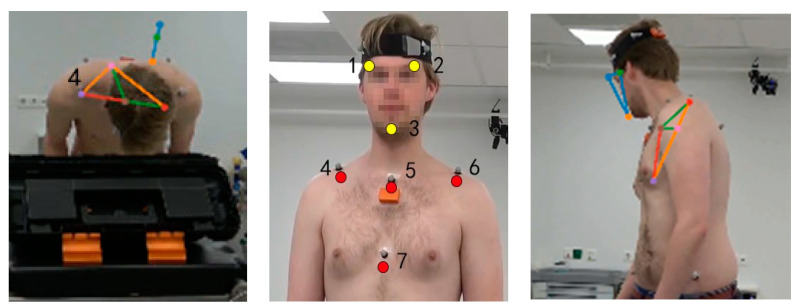
Example of tracking points bad detection. In the left figure, only tracking point 4 was tracked correctly, in the right figure, no tracking points were tracked correctly.

**Table 1 sensors-22-08342-t001:** Detail of simulated surgical task.

Task	Set-Up	Instructions
Picking up small objects, transferring them and putting them down	An operating table was placed in front of the participant. The height of the operating table was adjusted to the most comfortable position for participants. A box with tilted containers was placed on the operating table. The function of tilted containers is to block the vision of the participants so that they need to look down to see the forceps. Five small objects (bottle caps) were placed inside the box, two forceps were placed on the left table.	(1) Pick up both forceps on the left side of the table (one in each hand). (2) Use the forceps to grasp a bottle cap with the left hand from the left tilted container. (3) Transfer the object from the left-hand forceps to the right-hand forceps. (4) Put the bottle cap into the right tilted container. (5) Put the forceps down at the original position. (6) Get back to normal position.

**Table 2 sensors-22-08342-t002:** RMSE (mean ± SD), Range of motion difference (mean ± SD, in degrees), Limits of agreement and Relative ROM error between the IMU/Markerless method and the gold standard.

			Trunk FE	Trunk LB	Trunk AR	Neck FE	Neck LB	Neck AR
IMU method	SP-movements	RMSE	2.3 (1.3)	2.1 (0.9)	4.7 (1.7)	3.7 (2.2)	2.0 (1.0)	2.2 (1.1)
		ROM difference	**2.3** (2.1)	**2.2** (1.2)	**5.0** (2.9)	**5.4** (4.1)	0.2 (2.6)	0.4 (2.4)
		LOA	−1.9~6.4	−0.2~4.5	−0.6~10.6	−13.5~2.74	−4.9~5.3	−5.1~4.3
		Relative ROM error	0.035	0.053	0.070	0.11	0.0057	0.0058
	Simulated surgery task	RMSE	2.3 (1.1)	2.5 (1.2)	3.6 (1.8)	3.6 (2.2)	3.9 (2.0)	3.6 (2.1)
		ROM difference	**1.7** (1.9)	0.3 (2.8)	2.9 (4.2)	0.8 (4.1)	0.3 (2.8)	**2.2** (3.0)
		LOA	−2.1~5.4	−5.7~5.2	−5.3~11.1	−7.2~8.8	−5.2~5.8	−8.1~3.7
		Relative ROM error	0.043	0.0077	0.072	0.024	0.0084	0.048
Markerless method	SP-movements	RMSE	9.6 (12.5)	4.5 (4.0)	14.9 (10.1)	4.7 (3.0)	7.6 (3.8)	15.2 (8.2)
		ROM difference	**6.4** (7.1)	**5.5** (12.1)	11.7 (13.5)	2.9 (8.5)	2.8 (16.0)	**18.5** (10.4)
		LOA	−7.5~20.4	−29.2~18.2	−14.8~38.1	−13.6~19.5	−28.5~34.1	−1.9~38.8
		Relative ROM error	0.10	0.13	0.16	0.058	0.080	0.26
	Simulated surgery task	RMSE	5.5 (2.1)	5.6 (3.3)	8.7 (4.1)	6.1 (3.2)	7.0 (4.2)	7.3 (2.7)
		ROM difference	**8.3** (5.8)	4.8 (7.13)	3.6 (13.8)	10.3 (14.7)	**12.0** (11.1)	2.2 (9.9)
		LOA	−19.8~3.2	−19.3~9.6	−30.6~23.5	−39.4~18.9	−34.8~10.9	−24.3~20.0
		Relative ROM error	0.21	0.12	0.089	0.31	0.33	0.048

Bold values indicate a significant difference between the IMU/Markerless method and the gold standard.

**Table 3 sensors-22-08342-t003:** ICC (2,1) of ROM for the IMU/Markerless method.

			Trunk FE	Trunk LB	Trunk AR	Neck FE	Neck LB	Neck AR
IMU method	SP-movements	ICC (2,1)	**0.98**	**0.95**	**0.92**	**0.90**	**0.99**	**0.97**
		95% CI (*p* value)	0.74~1.00 (*p* < 0.001)	0.10~0.99 (*p* < 0.001)	0.03~0.98 (*p* < 0.001)	0.16~0.98 (*p* < 0.001)	0.96~1.00 (*p* < 0.001)	0.88~0.99 (*p* < 0.001)
	Simulated surgery task	ICC (2,1)	**0.96**	**0.97**	**0.95**	**0.80**	**0.96**	**0.95**
		95% CI (*p* value)	0.72~0.99 (*p* < 0.001)	0.88~0.99 (*p* < 0.001)	0.74~0.99 (*p* < 0.001)	0.39~0.95 (*p* < 0.01)	0.83~0.99 (*p* < 0.001)	0.73~0.99 (*p* < 0.001)
Markerless method	SP-movements	ICC (2,1)	**0.83**	**0.59**	**0.08**	**0.86**	**0.09**	**0.28**
		95% CI (*p* value)	0.25~0.96 (*p* < 0.001)	0.041~0.88 (*p* < 0.05)	−0.25~0.56 (*p* = 0.351)	0.57~0.96 (*p* < 0.001)	−0.61~0.67 (*p* = 0.408)	−0.10~0.72 (*p* < 0.05)
	Simulated surgery task	ICC (2,1)	**0.55**	**0.70**	**0.56**	0.31	**0.47**	0.42
		95% CI (*p* value)	−0.11~0.88 (*p* < 0.01)	0.18~0.92 (*p* < 0.01)	−0.06~0.87 (*p* < 0.05)	−0.19~0.75 (*p* = 0.122)	−0.11~0.83 (*p* < 0.05)	−0.27~0.82 (*p* = 0.1)

Bold values indicate statical significance.

## Data Availability

The data presented in this study are available on request from the corresponding author. The data are not publicly available due to ethical and privacy reasons.

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
