# Peer review of "Pilot Validation Study of Inertial Measurement Units and Markerless Methods for 3D Neck and Trunk Kinematics during a Simulated Surgery Task"

_sensors, 2022, doi:10.3390/s22218342_

Round 1
Reviewer 1 Report
Brief summary: The present work aimed at validating IMU and markerless methods through an optoelectronic system during a simulated surgery task. Results recommended the use of IMUs rather than markerless motion capture technologies.
Major comments:
The topic of this work is original, causing a probable interest in the reader. However, some changes are needed to improve the clarity of presentation and the comprehensibility of the study.
The introduction is well-written and the aim is clearly stated. However, the use of IMUs for the evaluation of spinal posture in different movements is not explored. Authors could add these references to make the introduction more exhaustive:
- Digo, E., Pierro, G., Pastorelli, S., & Gastaldi, L. (2019, June). Tilt-twist method using inertial sensors to assess spinal posture during gait. In International Conference on Robotics in Alpe-Adria Danube Region (pp. 384-392). Springer, Cham.
- Michaud, F., Lugrís, U., & Cuadrado, J. (2022). Determination of the 3D Human Spine Posture from Wearable Inertial Sensors and a Multibody Model of the Spine. Sensors, 22(13), 4796.
The section of materials and methods should be rearranged according to the following order: participants, materials (marker-based system, IMUs, and markerless system), study approach, and data analysis. How did you fix IMUs on the trunk? Moreover, I suggest creating a bulleted list for the statistical analysis for a better clarity. Finally, Bland-Altman plots should have a title indicating the trials to which they refer (SP movements or simulated surgery).
Results are deeply and consistently discussed.
Minor comments:
- Line 17: explain the acronym OR
- Line 67: replace “kg/m2” with “kg/m2”
- Line 82: explain the acronym SP
- Line 113: please, specify the measurement range also for the gyroscope and the magnetometer
Reviewer 2 Report
The wors seems very interesting and useful for further research and development of wearable health care. Authors are requested to address the following comments before accepting.
1. What is OR in the abstract? Application needs to be little more elaborated to motivate readers and other people in the area.
2. Four Xsens IMUs were used for the experiment. Pls include some key details of the device and data collection process like the device form factor, what controller they used, how the data was collected and stored before they transfer for further processing.
As it is claimed as better than bulky optoelectronics, what is the advantage in terms of size, wearablibilty, comfortability the user gets when uses this four IMUs system?
3. why four Xsens, was the same results can be achieved by three IMUs with better processing? is there any trade-off between number of devices vs processing efficiency/accuracy? what is 5 IMUs were used, would it more accurate?
4. How the collected data between the camera and IMUs are synchronized?
5. is there any feedback for the surgeon? How monitoring IMUs data will help surgeon to take preventive measures to reduce their risk.
6. Pls compare the results with some related work may be in table from literature review.
Round 2
Reviewer 2 Report
Accepted.